# Enhanced Circular Dichroism by F-Type Chiral Metal Nanostructures

Yuyuan Luo [1], Jin Liu [1,*], Haima Yang [2], Haishan Liu [1], Guohui Zeng [1] and Bo Huang [1]

[1] School of Electronic and Electrical Engineering, Shanghai University of Engineering Science, Shanghai 201620, China; m325121316@sues.edu.cn (Y.L.); 02050014@sues.edu.cn (H.L.); zenggh@sues.edu.cn (G.Z.); huangbosues@sues.edu.cn (B.H.)
[2] School of Optical-Electrical and Computer Engineering, University of Shanghai for Science and Technology, Shanghai 200093, China; snowyhm@usst.edu.cn
* Correspondence: liujin@sues.edu.cn; Tel.: +86-137-6193-6139

**Abstract:** Circular dichroism (CD) effects have broad applications in fields including biophysical analysis, analytical chemistry, nanoscale imaging, and nanosensor design. Herein, a novel design of a tilted F-type chiral metal nanostructure composed of circular nanoholes with varying radii has been proposed to achieve remarkable CD effects, and the results demonstrate the generation of a significant current oscillation at the sharp edges where the nanoholes overlap under circularly polarized light, resulting in a strong CD effect. The CD effect can reach up to 7.5%. Furthermore, spectral modulation of the resonant wavelength can be achieved by adjusting the structural parameters, which enhances the tunability of the structure. Overall, these results provide theoretical or practical guidance for enhancing the circular dichroism signal strength of chiral metal nanostructures and designing new types of two-dimensional chiral structures.

**Keywords:** chiral; circular dichroism; left circularly polarized; right circularly polarized; localized surface plasmon

## 1. Introduction

Chirality is a fundamental quality of an object that cannot be transferred to its mirror image via spatial operations, such as rotation or translation [1]. It can be classified as intrinsic chirality and extrinsic chirality. Intrinsic chirality primarily refers to the symmetry breaking within the material itself, while extrinsic chirality refers to the symmetry breaking of the incident electromagnetic field [2,3]. When the size of metal structures with intrinsic chirality attributes is reduced to the micro- and nanoscale, their interaction with left circularly polarized (LCP) and right circularly polarized (RCP) light induces specific electromagnetic responses [4]. The circular dichroism effect refers to the fact that, when LCP and RCP light are incident perpendicularly on metal nanostructures with intrinsic chirality, a difference in the transmission spectra can be calculated by the difference value between the transmittance of the LCP and RCP light [5]. The CD is strongly linked to the absorption loss of the metallic material in the micro nanostructure, which is represented by the imaginary component of its refractive index. Metal nanostructures, in general, exhibit higher chiral properties compared to natural chiral molecules and have extensive applications in the design of nanosensors based on surface plasmon resonance (SPR) effects [6], nanomanipulation [7], near-field trapping technology [8], nanoscale imaging [9], and biosensing [10].

Circular dichroism is widely used in organic [11], biochemistry [12], and medicinal chemistry [13]. Hence, various metal chiral nanostructures have been proposed to explore the physical mechanisms underlying the CD effect in metal chiral structures that encompass two-dimensional (2D) flat structures and three-dimensional (3D) double- or multi-layered chiral nanostructures. By irradiating the surface of metal nanostructures with circularly

polarized light, the CD effect can be achieved, and its underlying reasons have been explored. The CD characteristics have been achieved through theoretical calculations, and researchers have successfully fabricated metal chiral nanostructures and detected their CD effects through experiments. In 1811, circular dichroism and circular birefringence were first discovered in quartz materials by scholars such as F.J.D. Arago. In recent years, to enhance the CD effects, various 3D nanostructures with double or multiple layers have been designed. For instance, Justyna et al. [14] designed a chiral metamaterial with a spiral gold structure, which exhibits significant differences in transmittance between LCP and RCP light upon irradiation, resulting in strong CD effects. By adjusting the number of spiral layers, they can control the range of polarization wavelengths and obtain responses across different wavebands. CD effects in 3D chiral nanostructures arise from the coupling between layers, and strong near-field coupling between two-layered nanostructures can also lead to CD effects. For instance, in 2006, Zheludev's group [15] designed a chiral nanostructure featuring a double-layered metal rose-shaped structure, which demonstrates remarkable optical activity. However, fabricating 3D metal chiral nanostructures is more challenging and complex compared to 2D chiral metal nanostructures. In contrast, 2D chiral metal nanostructures have fewer technical requirements and are simpler to fabricate, also holding great potential for various applications in optical devices and thus considered more suitable for their utilization. Numerous studies have been conducted on the CD effects of 2D metal nanostructures, including planar chiral quasicrystals [16], asymmetric split-ring structures [17], L-shaped particle disk combinations [18], G-shaped structures [19], planar heptamers [20], etc. The CD effects in these structures arise from the strong local surface plasmon (LSP) resonance that occurs at the metal–dielectric interface. Furthermore, non-chiral metal nanostructures have been found capable of acquiring chiral traits through tilted incident methods that induce structural CD effects. Factors including the geometric shape, size, material, and surrounding environment all significantly impact the CD effects observed in metal chiral nanostructures. This approach is endowed with considerable potential for modulating the CD effects of such structures.

At present, focused ion beam, electron beam lithography, SP (surface plasmon) super-resolution lithography, and other technologies have been rapidly developed. For instance, when an electron beam strikes a metal film, a series of modifications, including the electron beam spot adjustment and the electron beam deflection correction, result in the creation of a nanoscale pattern on the substrate sheet, which, as has been experimentally proven, facilitates the preparation of more complex nanostructured optical devices [21]. Therefore, in order to achieve a more obvious circular dichroism effect, inclined F-type metal chiral nanostructures were hereby designed, which comprise two circular nanoholes with varying radii. Meanwhile, the transmission spectra and CD characterization of this structure were theoretically investigated using the finite element method (FEM). The results indicate that, when subjected to vertical irradiation by LCP and RCP light, strong electronic oscillations occur on the metal surface of the F-type composite structure. Significant current oscillatory coupling occurs at the intersection of circular nanopores. This leads to a more significant transmission peak of the nanostructure at the resonance wavelength, thereby bringing about two distinct CD signals. In addition, it was explored that changes in the structural parameters of the F-type chiral metal nanostructures would have different effects on the generation of CD signals from the structures, and flexible modulation of the optical effect is achieved by varying the gap parameter between two neighboring circular nanopores; the tilt angle of the composite structure, the radius of the nanopore, and the nanostructure perimeter were altered to increase the utility of the F-type nanostructures.

## 2. Materials and Methods

The F-type metal chiral nanostructure proposed in this study is depicted in Figure 1. Figure 1a depicts the unit structure schematically, which comprises four inclined non-chiral gold nanoholes and four inclined adjacent non-chiral gold nanoholes with radii $r_1$, $r_2$, thickness $t$, spacing $a$ between two adjacent nanopores and the inclination angle $\theta$. The

electromagnetic characteristic parameters of gold (Au) are determined using the Drude model [22]. Figure 1b represents a 3D structural design of the nanostructure, demonstrating periodicity along the x and y directions. The periodic structure is comprised of unit structures periodically arranged in the x-o-y plane, which is theoretically infinitely extended. The geometric dimensions of each unit structure correspond to their electromagnetic properties, and all structures were hereby designed on a 200 nm thick silicon dioxide substrate having a refractive index of 1.45.

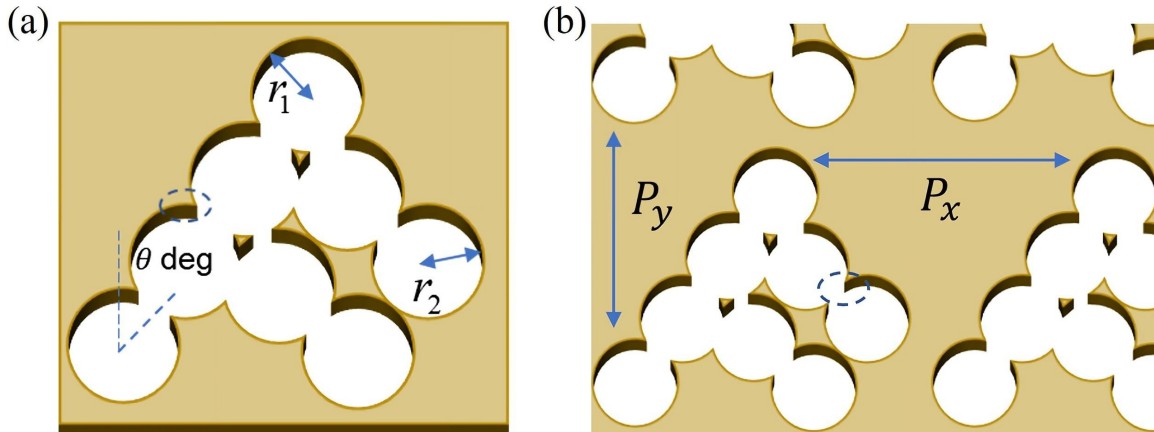

**Figure 1.** (**a**) The unit schematic of F-type metallic hand-shaped nanostructures. (**b**) The schematic for an array of F-type metallic hand-shaped nanostructures.

Herein, the transmission spectra, CD spectra, and surface current charge distributions of an F-type metal chiral nanostructure at its resonance wavelength were calculated using the COMSOL Multiphysics software. Circularly polarized light was set as the input and output ports of the nanostructure during the calculation process. The input light, composed of LCP and RCP light, is incident vertically along the z-axis. The three electric field components of RCP light are $E_x = 1$, $E_y = i$, $E_z = 0$, while those of LCP light are $E_x = 1$, $E_y = -i$, $E_z = 0$. To simulate periodic arrays of F-type metal chiral nanostructures with periodicity, periodic boundary conditions were applied along the x and y directions. Perfect matching layers were also introduced to ensure the complete absorption of outgoing light. Generally, when chiral metal nanostructures interact with light, CD effects will be generated through the simultaneous activation of electric and magnetic dipole modes, resulting in the production of dipole moments represented by electric dipole moments (1) and magnetic dipole moments (2), as reported in [23]:

$$\tilde{P}_e = \tilde{\alpha}\tilde{E} - i\tilde{G}\tilde{B}, \tag{1}$$

$$\tilde{P}_m = \tilde{\chi}\tilde{B} - i\tilde{G}\tilde{B}, \tag{2}$$

where $\tilde{\alpha}$ represents the electric polarizability, $\tilde{\chi}$ the magnetic permeability, $G$ the mixed electric–magnetic dipole polarizability, $E$ and $B$ the local fields on the metal surface [24]. Under the excited conditions of LCP and RCP, the absorption $A^{\pm}$ intensity of the chiral metal nanostructure can be expressed as [25]:

$$A^{\pm} = \frac{\omega}{2} Im(\tilde{E} \cdot \tilde{P}_e + \tilde{B}^* \cdot \tilde{P}_m), \tag{3}$$

according to the optical characteristics of the chiral structures, the CD can be expressed as

$$CD = A_{++} - A_{--}, \tag{4}$$

where $A_{++}$ is the absorption rate under RCP irradiation and $A_{--}$, the absorption rate under LCP irradiation. Meanwhile, the absorption intensity of the metal chiral nanostructures can again be expressed as

$$A = 1 - T - R, \qquad (5)$$

where $T$ refers to the transmittance and $R$ the reflectance. Therefore, when ignoring the reflection, CD can also be expressed as

$$CD = T_{++} - T_{--}, \qquad (6)$$

where $T_{++}$ is the transmittance under RCP light irradiation and $T_{--}$ is the transmittance under LCP irradiation.

## 3. Results and Discussion

### 3.1. Spectral Properties of Circular Dichroism of Metal Nanostructures

To examine the resonance modes of chiral response in nanostructures, the transmission and CD spectra of the nanostructure for RCP and LCP light incidences are illustrated in Figure 2. The spectrum range spans from 620 nm to 780 nm. During the simulation process, the structural parameters of the F-shaped nanostructure were set with a periodicity of $P_x = P_y = 820$ nm, nanoholes of $r_1 = 100$ nm, $r_2 = 110$ nm, and an angle of inclination of $\theta = 30°$. Figure 2a presents the CD spectrum of the F-shaped metal chiral nanostructure, while Figure 2b exhibits the transmission spectrum of the same structure. At the resonant wavelength of $\lambda_1 = 650$ nm, the transmittance curves $T_{++}$, $T_{--}$ exhibit a significant resonance peak, resulting in a pronounced difference and a prominent CD signal, referred to as Mode 1. At approximately $\lambda_2 = 733$ nm, another resonance peak was observed, leading to a weaker CD signal named Mode 2. Furthermore, in the transmission spectrum, the transmittance curve $T_{--}$ corresponding to Mode 1 surpasses the curve $T_{++}$, indicating negative dichroism in the CD spectrum. Conversely, for Mode 2, the transmittance curve $T_{--}$ is lower than curve $T_{++}$, resulting in positive dichroism in the CD spectrum.

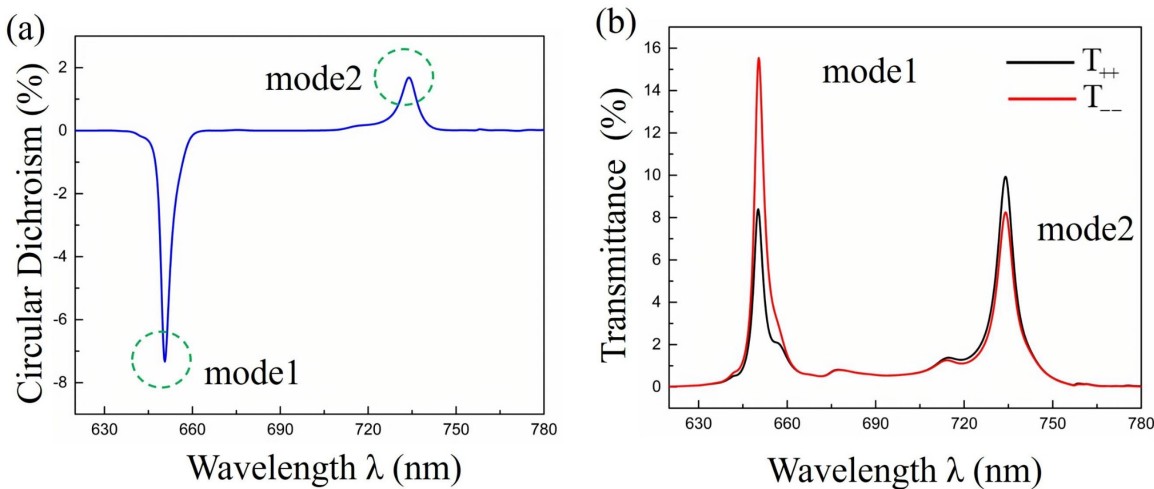

**Figure 2.** The transmission spectrum and CD spectrum of F−type metallic nanostructure: (**a**) CD spectrum; (**b**) transmission spectrum.

To elucidate the mechanism behind the CD effect in the nanostructure, the current density distribution at the resonance wavelengths and the electric field intensity in the Z-direction were hereby computed and analyzed. Figure 3 illustrates the current density for Modes 1 and 2 under the excitation of LCP and RCP. Figure 3a presents the current density distribution in the Z-direction of the electric field for Mode 1 under RCP and LCP excitation, respectively. In Mode 1, $\lambda_1 = 650$ nm, the primary current runs from the sharp corner of the nanohole towards both sides, causing the accumulation of charges at the sharp

corners, which can be equated to two dipole moments. Under LCP and RCP excitation, the two electric dipoles exhibit strongly localized surface plasmon (LSP) resonances, which enhances the electric field intensity in the surrounding region, leading to transmission peaks. Due to differences in intensity and direction of the oscillation of the dipole moments, the current density at the sharp corner of the nanohole under LCP excitation is significantly higher under RCP excitation, thus leading to stronger charge accumulation and enhanced resonance strength. Therefore, the transmission intensity under LCP excitation is greater than that under RCP excitation. In this case, the sign of the CD value in Mode 1 is negative, and the structure has negative dichroism.

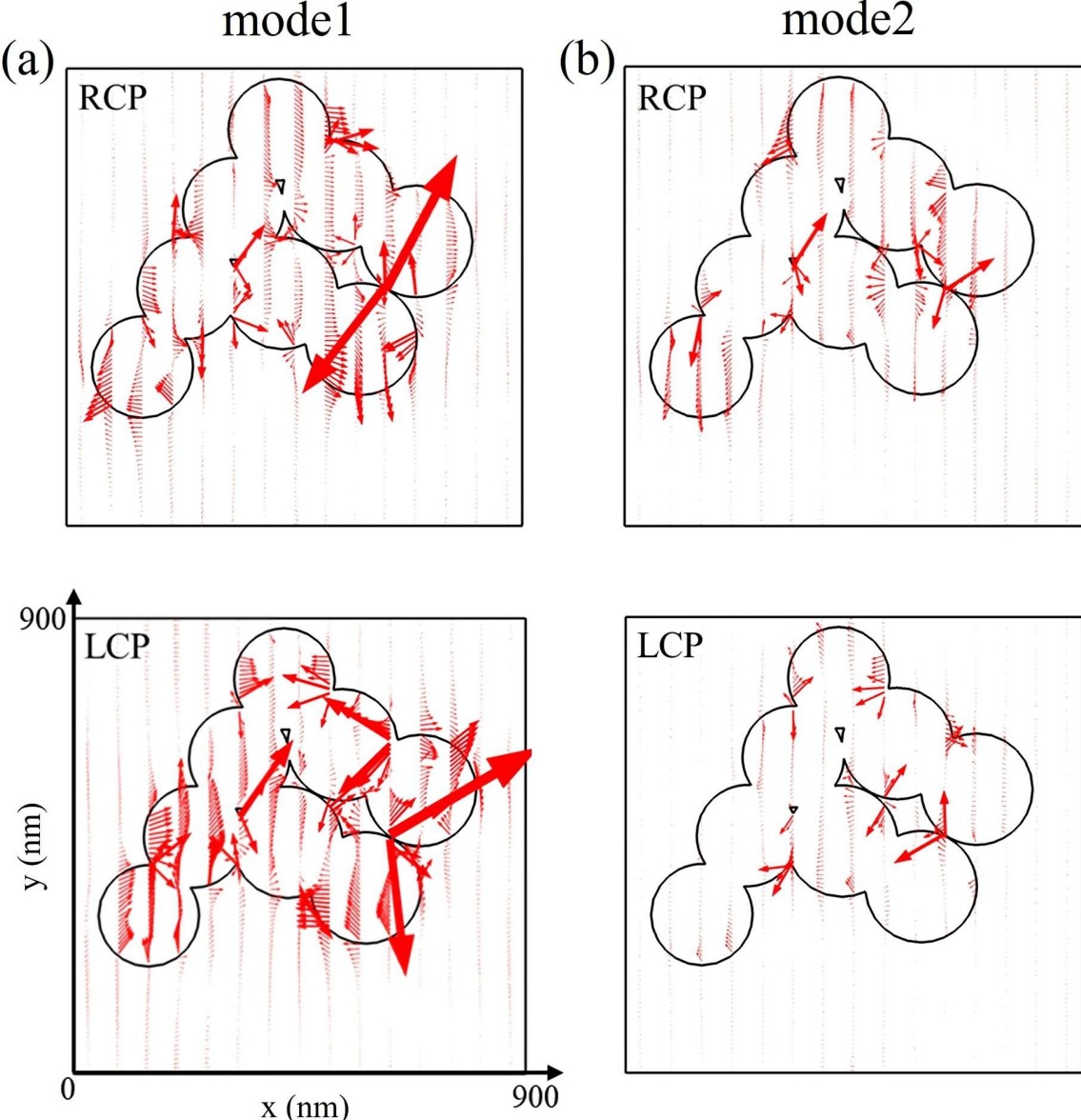

**Figure 3.** The figure shows the current density distribution at resonance (**a**) the distribution of the RCP and LCP irradiated current density (Mode 1); (**b**) the distribution of the RCP and LCP irradiated current density (Mode 2).

Figure 3b illustrates the current density distribution for Mode 2 under RCP and LCP excitations, respectively, where it can be observed that Mode 2 exhibits different oscillation intensities and directions compared to Mode 1. The primary current flows to both sides from the two sharp corners, forming a twisted four-dipole current oscillation and an electric

four-dipole resonance mode, thereby resulting in a transmission peak. Under the excitation of LCP and RCP, the current oscillation intensity under the excitation of RCP is higher than that under the excitation of LCP, which reduces the absorption rate and increases the transmittance, leading to the generation of the CD signal in Mode 2. Therefore, the sign of the CD value in Mode 2 is positive, and the structure has positive dichroism. Moreover, it is worth noting that the oscillation intensity difference produced by Mode 2 is relatively smaller than that produced by Mode 1, making the CD effect of Mode 2 weaker than that of Mode 1.

### 3.2. Effects of Metal Nanostructure Parameters on Chiral Properties

To investigate the influence of nanostructure parameters on the CD effect, the gap size of the nanopore of the F-type nanostructures, a tilt angle $\theta$, nanopore radius $r_1, r_2$, and period $P$ were sequentially adjusted based on each parameter nanopore gap $a = -20$ nm, tilt angle $\theta = 30°$, nanopore radius $r_1 = 100$ nm, $r_2 = 110$ nm, and period $P_x = P_y = 820$ nm, as shown in Figure 1. In the case of the adjustment of one of the parameters, the other parameters remain unchanged. Figure 4a shows the cell structure diagram when $a = -20$ nm (intersecting nanopore), $a = 0$ nm (tangent nanopore), and $a = 20$ nm (separated nanopore), and Figure 4b presents the CD spectra of the metal nanostructures at the gap $a = -20$ nm, 0 nm, 20 nm.

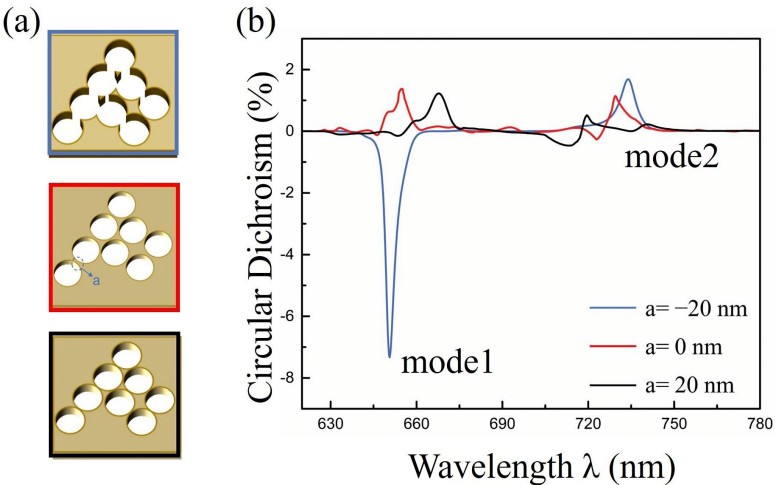

**Figure 4.** (**a**) The unit schematic of metal nanostructures with different gaps. (**b**) CD spectra of metallic nanostructures with different gaps.

Figure 5 illustrates the distribution of intensity of the electrical field for the metal nanostructure. The simulation results reveal that, as the nanohole spacing increases, the peak value of the CD spectra decreases progressively, and Mode 1 shifts towards longer wavelengths with a change in its peak value symbol. The minor red shift of the peak value is attributed to the closer interaction distance between adjacent nanoholes for a smaller spacing and a constant period. Such interference prevents sufficient interactions between molecules and results in a red shift in the surface resonance of the nanostructure. The change in the symbol of the peak, on the other hand, is attributed to the fact that the transmission intensity of the LCP excitation is greater than that of the RCP excitation when $a = -20$ nm, while the opposite is true for $a = 0$ nm and $a = 20$ nm, where the transmission intensity of RCP excitation is greater than that of LCP excitation. A clearer understanding of this phenomenon can be obtained from figures of the electric field intensity distribution map. For instance, Figure 5a–f display electric field intensity distributions in three chiral structures in Mode 1 under both RCP and LCP excitations. The effective charge oscillation in all three chiral nanostructures predominantly occurs at the apex of intersecting nanoholes in Mode 1. The absence of this apex leads to a decrease in the effective charge oscillation

and diminishes the electric field intensity. Moreover, Mode 2 demonstrates a similar decline in the CD effect with the increase in the nanohole spacing but exhibits a huge blue shift in the resonance wavelength. As the distance between adjacent nanoholes increases gradually due to the rise in $a$, the effective oscillation ratio between nanoholes decreases so that the localized surface plasmon resonance (LSP) between them is reduced. Consequently, the resonance wavelength of light shifts to blue, and the CD mode also shifts in the same direction. Under RCP and LCP excitations, Figure 5g–l illustrate the electric field intensity patterns of the three chiral structures in Mode 2, where it can be observed that the electric field intensity increases at the sharp corners, thus increasing the effective ratio of oscillating currents and resulting in a significant CD signal increase. By comparing the electric field strength distribution maps for the three chiral structures under RCP excitation and LCP in Mode 2, the study found an easy association with previous studies.

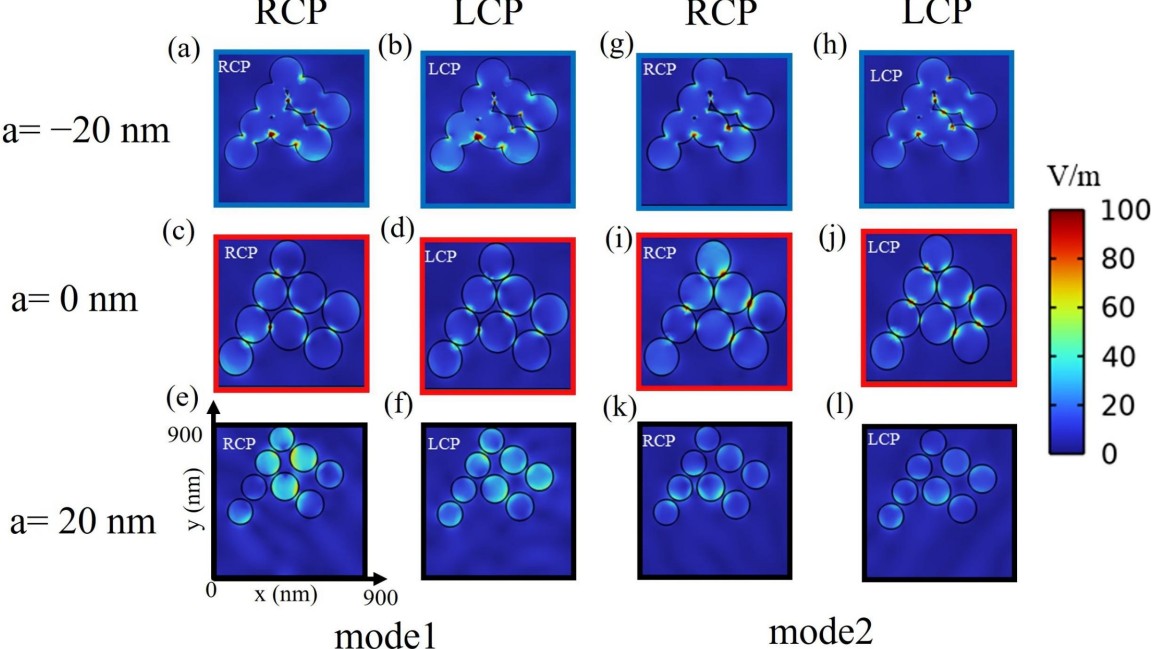

**Figure 5.** The electric field intensity for the metal nanostructures with varying gaps (a = −20 nm (blue rectangle), a = 0 (red rectangle), and a = 20 nm (black rectangle)). (**a–f**) Electric field intensity in Mode 1. (**g–l**) Electric field intensity in Mode 2.

In Figure 6a, the nanostructure parameters remain fixed with a period of $P_x = P_y = 820$ nm and a nanohole diameter of $r_1 = 100$ nm, $r_2 = 110$ nm. The absolute value of the CD effect was taken for accurate analysis, as shown in Figure 6b. To assess the impact of tilt angle on the F-shaped nanostructure, Mode 1 was divided into two parts for discussion. The first reveals that the CD value undergoes blue shift and intensity increase with the increase in tilt angle from $\theta = 15°$ to $\theta = 30°$. As the tilt angle of the nanoholes increases from 15° to 30°, the effective oscillating current on the nanostructure $\theta = 30°$. This increased coupling between the nanoholes and reduction in resonant path distance result in a significant blue shift in the CD peak as well as an increase in CD intensity. When $\theta = 45°$, the spatial symmetry of the nanostructure is at its best and the chiral property of the structure is at its lowest point. Therefore, in Mode 1, the CD value is at its minimum for $\theta = 45°$. Furthermore, as the tilt angle gradually increases from 45° to 75°, the spatial symmetry of the nanostructure is progressively worsened, resulting in an increased electromagnetic coupling effect between the nanoholes. This coupling effect further enhances the CD effect of the structure. Additionally, the tilt angle is $\theta = 60°$. Due to the change in tilt angle, the transmitted intensity is greater during RCP excitation than it is during LCP excitation, indicating that the CD value is positive. In Mode 2, a slight

red shift is observed in the CD mode, and the corresponding CD value starts to decrease gradually as the tilt angle of the nanoholes increases from 45° to 75°. Furthermore, in Mode 2, the effective current oscillation is distributed within the F-shaped nanostructure at the intersection of the two intersecting nanoholes. The chirality of the nanostructures decreases slowly with the increase in the tilt angle. The current density between the nanoholes decreases gradually, resulting in a weakening of the charge accumulation at the nanohole intersection. This, in turn, leads to a gradual decrease in the transmittance strength within the structure, causing a decrease in the CD value. Meanwhile, given that the change in the tilt angle makes the RCP excitation under the transmittance intensity greater than the LCP excitation of the transmittance intensity, the circular dichroic effect changes the sign to positive dichroic when the tilt angle is between 30 and 45 degrees.

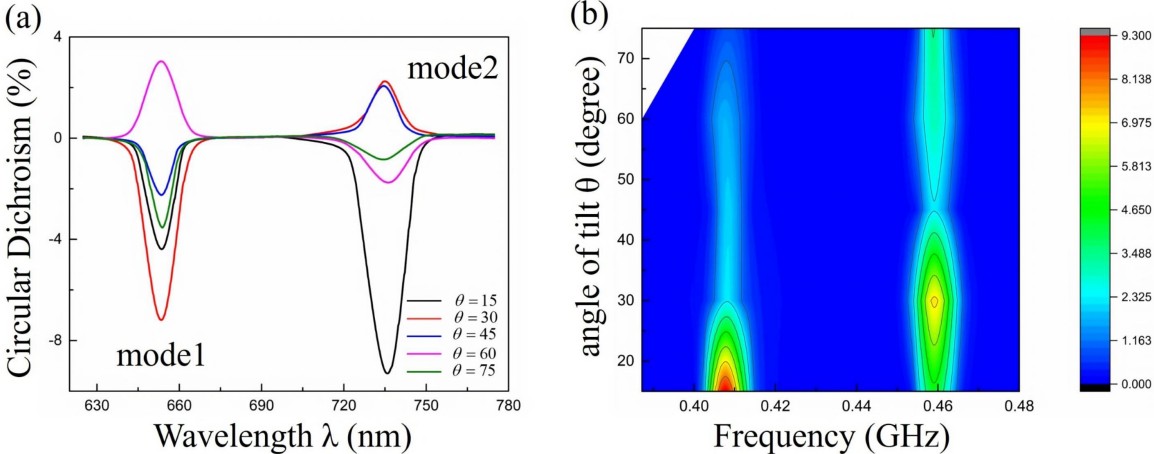

**Figure 6.** (**a**) CD spectra of F−type metal chiral nanostructures at different tilt angles. (**b**) Relationship between CD values and period and frequency ($f = c/\lambda$).

Figure 7 shows the effect of different nanopore radii $r_1, r_2$ on the F-shaped nanostructures with other parameters fixed as $P_x = P_y = 820$ nm, $\theta = 30°$. First, to simulate each of these six distinct chiral modes, $r_2$ was set to 90 nm, 100 nm, and 110 nm, respectively, when $r_1$ was fixed at 100 nm, as shown in Figure 7a. Then, as shown in Figure 7b, $r_1$ was set to 90 nm, 100 nm, and 110 nm, respectively, when $r_2$ was fixed at 110 nm. The effective current oscillation of the nanostructures in Mode 1 looks like the oscillation of two electric dipoles. When the nanopore radius $r_2$ decreases from 110 nm to 100 nm and then to 90 nm, the position of the CD mode is almost unchanged, but the CD value experiences a rapid decrease when $r_2$ decreases from 110 nm to 100 nm. This is because, when $r_2 = 100$ nm, the asymmetry of the chiral structure is destroyed and the chirality of the nanostructure is weakened, which leads to a decrease in the transmittance under LCP irradiation. However, the transmittance under RCP irradiation remains unchanged, and, therefore, the CD value decreases and becomes positive dichroism. As $r_2$ decreases to 90 nm, the coupling between the nanopores decreases, making the effective current oscillation intensity at the sharp corners and transmission intensity smaller. At this time, the current oscillation distance between the nanopores is almost the same, so the position of the CD modes is almost the same and the value of CD decreases. When $r_2 = 90$ nm, the chirality of the nanostructures becomes stronger compared to $r_2 = 100$ nm, but the smaller $r_2$ makes both the coupling between the nanopores and the transmission intensity smaller, so the CD value increases only slightly. When $r_1$ decreases from 110 nm to 100 nm, the chirality of the nanostructures is enhanced and the coupling between the nanopores is strengthened, which makes the transmission intensity under LCP irradiation larger, and thus the CD intensity becomes larger but presents negative dichroism. The coupling between the nanopores weakens when $r_1 = 90$ nm, lowering the transmission intensity under RCP irradiation and lowering the CD intensity. The CD peak will blue shift as a result of the decreasing resonance distance

between the nanopores. In Mode 2, the nanostructure is a twisted four-dipole oscillation. Therefore, when $r_2$ decreases, the resonance intensity between nanopores on the short side decreases correspondingly, while the resonance intensity between nanopores on the long side remains unchanged. The effective ratio of the oscillating current between the nanoholes decreases, the resonance wavelength is blue-shifted, and the CD value decreases. As in Mode 1, the CD value is slightly larger than $r_2 = 100$ nm when $r_2 = 90$ nm. The resonance intensity of the nanopore at the long edge decreases as $r_1$ shrinks from 110 nm to 100 nm, but that of the resonance intensity of the nanopore at the short edge remains unchanged. As a result, during RCP irradiation, the transmission transmittance peaks rise, which, however, stay the same under LCP irradiation. The CD effects consequently increase, and a decrease in the resonance gap between the nanopores causes the CD peaks to be blue-shifted. The aforementioned proof shows that the CD impact of nanostructures is strongly influenced by the nanopore radius.

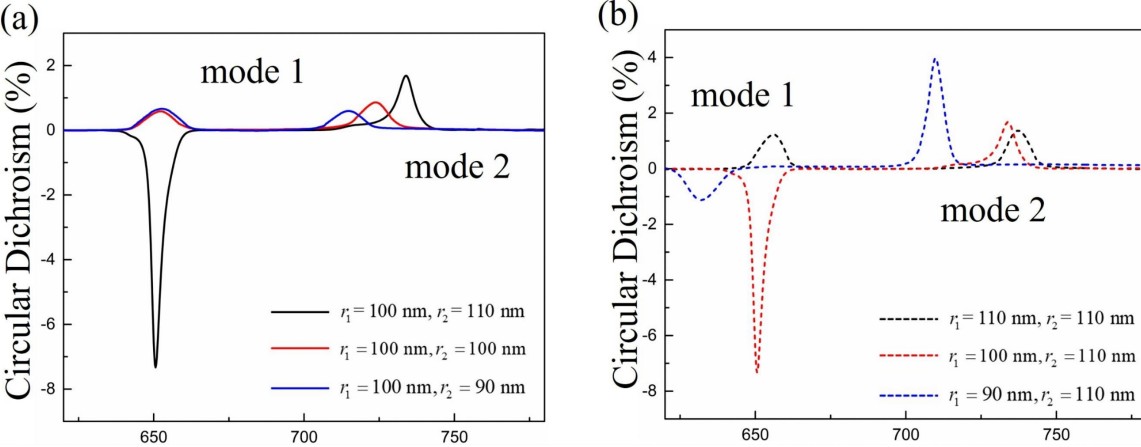

**Figure 7.** (**a**) CD spectra of $r_1 = 100$ nm, $r_2 = 110$ nm, 100 nm, 90 nm; (**b**) CD spectra of $r_1 = 110$ nm, 100 nm, 90 nm, $r_2 = 110$ nm.

Finally, it is also effective to achieve tunability by changing the period of the chiral structure on the same chip. In addition to the thickness of the chiral layer or the refractive index of adjacent media, adjusting the periodic structure of the chiral structure on the same chip is also an effective approach to achieving tunability [26]. Due to the nanostructure array set from 810 nm to 890 nm, five F-type metals with the following characteristics in the wavelength range of 720 nm to 900 nm were hereby fabricated to more accurately analyze the tuning effect of the array period on the nanostructure, including chiral nanostructure array ($r_1 = 100$ nm, $r_2 = 110$ nm, $\theta = 30°$), which has the same cell and different array periods ($P = 810$ nm, 830 nm, 850 nm, 870 nm, 890 nm). Figure 8a,b depict the CD effect spectra for various periods and the fluctuation of transmittance with period and frequency ($f = c/\lambda$), respectively. A rather noticeable overall red shift of the transmission peak can be observed in the case of an array period raising from 810 nm to 890 nm, as shown in Figure 8b, resulting in a greater electronic resonance distance and a red shift of the resonance wavelength since the breadth between two adjacent arrays of F-type metal chiral nanostructures grows as the period gradually increases. Two scenarios allow for discussion of its CD value: first, when P shifts from 810 nm to 830 nm, the CD value is negative and gradually drops. The CD value is positive and slowly falls when P shifts from 850 to 890 nm. The resonant mode of an array structure is closely related to its periodicity and the incident wavelength of incoming light, as described by equation $\lambda_{SPP} = \frac{P}{\sqrt{i^2+j^2}}\left(\sqrt{\frac{\xi_d\xi_m(\omega)}{\xi_d+\xi_m(\omega)}}\right)$ [27], where P represents the structural periodicity, $i$ and $j$, integers, $\xi_d$, the dielectric constant of the dielectric, and $\xi_m(\omega)$, the complex dielectric function of the metal. Normal illumination generates a large transmission peak and the strongest optical chiral response when the periodicity matches the wavelength of SPPs propagating on the metal–dielectric interface.

As depicted in Figure 8a, when $P = \lambda$, the SPPs generated on the F-type metal chiral nanostructure surface reach the best match to the array periodicity P and thus the largest CD value, perfectly illustrating how nanostructure modification of the period affects the CD effect.

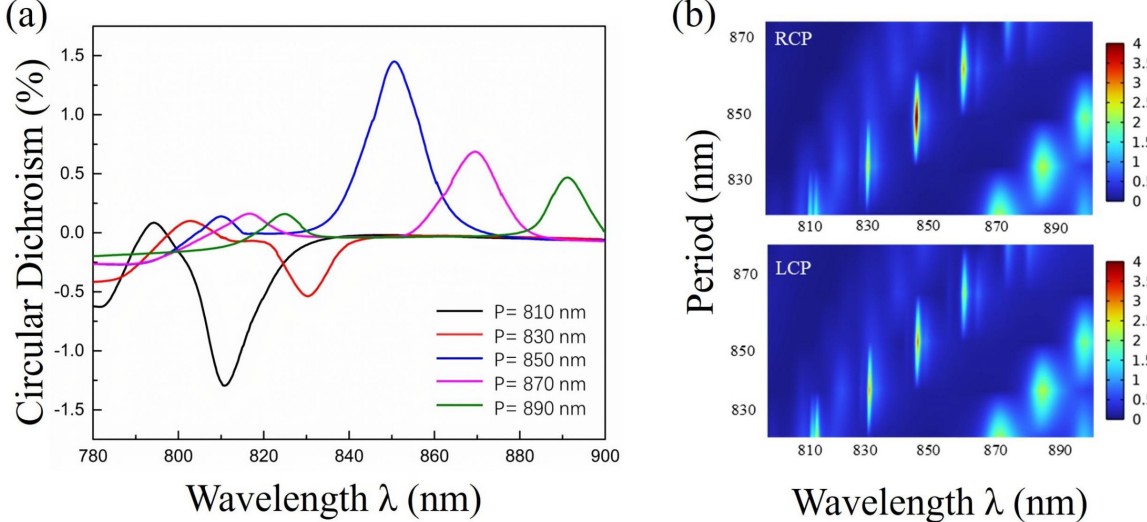

**Figure 8.** (**a**) CD spectra of arrays with the same unit cell but different periodicities. (**b**) Transmittance relation to period and frequency.

Finally, Table 1 shows the comparison of CD properties based on metal chiral nanostructures in visible light and near-infrared bands in recent years (converting the CD value to the difference in transmittance), revealing the excellent chiral properties of the F-type metal chiral nanostructures studied in this paper.

**Table 1.** Comparing the CD effects of different designs.

| Chiral Nanostructures | Optical Chirality | Wavelength Range (nm) | CD Value (%) | Reference |
|---|---|---|---|---|
| Crossed nanorods with nanowire | $CD = T_{++} - T_{--}$ | 600–1000 | 0.18/−0.15 | [28] |
| Twisted Z-shaped nanostructure(TZN) | $CD = T_{++} - T_{--}$ | 500–1000 | 0.868 | [29] |
| Nanoholes in mental film/Tilted nanorods | $CD = T_{++} - T_{--}$ | 600–2000 | 0.12 | [30] |
| Nanowire/G-type nanostructure | $CD = T_{++} - T_{--}$ | 300–6000 | −0.309/0.44 | [31] |
| Gold bilayer slit array/rectangular holes | $CD = T_{++} - T_{--}$ | 400–3000 | 0.167 | [32] |
| Nano slits milled in gold layer | $CD = T_{++} - T_{--}$ | 600–800 | 0.95 | [33] |
| F-type metal nanostructure | $CD = T_{++} - T_{--}$ | 620–780 | −7.5/1.5 | this work |

## 4. Conclusions

Herein, the finite element approach was employed to study the transmission and CD characteristics of a tunable F-type metal chiral nanostructure consisting of two circular nanoholes with different radii. Under circularly polarized light excitation, two CD modes and resonant modes of four dipoles and octupoles are observed at the resonant wavelength, resulting in CD effects of 7.5% and 1.5%, respectively. Meanwhile, by analyzing the electric field and current density distributions at the resonant wavelength, the fundamental physical mechanism behind the CD effect of the metal chiral nanostructure is revealed. The study also demonstrates that the CD effect of the electric dipole mode is five times stronger than that of the four-dipole mode, that the formation of CD properties is attributed to the resonant coupling of the acute angles at the intersections of circular nanopores, and

that the CD properties of the nanostructured arrays strongly depend on the structural parameters' nanopore interstitial gap a, tilt angle thea, nanopore radius r, and the periodic parameter P. As these angles disappear, the CD signals of the two CD modes weaken and correspondingly experience a slight red shift as well as a significant blue shift. In Mode 1, the CD effect is weakest when $\theta = 45°$, while, in Mode 2, the CD effect gradually decreases as the angle of inclination decreases. When $r_1 = 100$ nm and $r_2 = 110$ nm, the nanostructure presents the best CD effect. Moreover, the chiral effect is the strongest when the period is matched with the SPP wavelength. Additionally, the flexible modulation of the CD effect is realized through the adjustment of each parameter. Overall, these findings forge a theoretical foundation and guidance for enhancing the CD signal intensity of 2D metal chiral nanostructures and designing innovative structures.

**Author Contributions:** Conceptualization, Y.L.; Data curation, H.Y.; Investigation, Y.L. and B.H.; Methodology, J.L.; Project administration, J.L.; Supervision, G.Z. and H.L. All authors have read and agreed to the published version of the manuscript.

**Funding:** This research was funded by the Shanghai Science and Technology Innovation Action Plan (22S31903700, 21S31904200); National Nature Science Foundation of China (617011296, U1831133); Natural Science Foundation of Shanghai (17ZR443500).

**Institutional Review Board Statement:** Not applicable.

**Informed Consent Statement:** Not applicable.

**Data Availability Statement:** Not applicable.

**Conflicts of Interest:** The authors declare no conflict of interest.

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
