# Peer review of "Enhanced Circular Dichroism by F-Type Chiral Metal Nanostructures"

_photonics, doi:10.3390/photonics10091028_

Round 1

Reviewer 1 Report

Report in the attached file

See attached file

Reviewer 2 Report

This paper presented a theoretical design of a tilted F-type chiral metal nanostructure composed of circular nanoholes to achieve the efficient CD effects. The results is of interest but not detailed and comprehensive. And thus, these revised suggestions should be considered before being published.

1  Why F-type rather than Q-type, R-type or any other type was selected?

2  In introduction section, more quantitative descriptions should be added, especially for CD value.

3  Please check Figure 1, the symbol r1, r2 and θ should be appeared in Figure 1(a), according to the text description in line 76-80. Besides, Figure 1(a) and 1(b) are similar, and then Figure 1(b) is unnecessary.

4  All symbols in Eq.1-6 should be explained or defined, such as ,, , , ++ and --.

5  The nanohole diameter is be represented as d rather than r. And, tables should be used to present simulation parameters.

6  Please check the units of X-label, Y-label and your simulation results in all of your figures.

7  The influences of nanohole (diameter, spacing, configuration and number), periodicity and wavelenth on CD value and light intensity should be investigated comprehensively.

8  The comparison with other literatures, especially similar structures, should be added.

Minor editing of English language required.

Reviewer 3 Report

The study explores the applications of Circular Dichroism (CD) effects in various fields, proposing a new design for a tilted F-type chiral metal nanostructure with circular nanoholes of varying radii. Under circularly polarized light, the nanostructure generates significant current oscillation at the nanohole edges, resulting in a remarkable CD effect of up to 7.5%. The resonant wavelength can be modulated by adjusting structural parameters, enhancing the structure's practicality. These findings offer valuable insights into enhancing the circular dichroism signal strength of chiral metal nanostructures and designing novel two-dimensional chiral structures. However, some issues need to be addressed for the scientific soundness of the paper.

1.       For the first part of the paper, the author needs to clarify the definition of CD and the simulation results. Equation 4 writes CD = A++ −A--. And the author further writes in Equation 5 A = 1T R, and Equation 6 CD = T++ −T--. Is Equation 6 supposed to be CD = T-- −T++ based on the author’s definition? It would also help to specify if T++ represents the transmission for LCP and RCP, which is related to the next point I want to raise.

2.       For mode 1, T++ (LCP) is lower than T—(RCP), which is explained by the higher current density under LCP than RCP as stated by the author. The mechanism is right, but the figure caption is confusing. Figure 3 labels (a) for RCP and (b) for LCP. Clearly (a) shows higher current density. Can the author double-check it again? If it is not a simple mislabeling, the author should make more discussion about the simulated current density.

3.       How is the angle for a large CD related to the surface plasmon resonance angle of metal? It would offer a better understanding if there were a quantitative analysis.

4.       It is surprising to see that angle dependence is not monotonic. This indicates that the contributions from different spatial locations to the total electric field vary. It would be helpful to include the map of electric field intensity for different tilt angles.

5.       There are some minor grammatic errors. For example, on line 41, it would be ‘effects‘ instead of ‘affects’. There are other places with typos that need proofreading.

Round 2

Reviewer 1 Report

In the attachment

In the attachment

Reviewer 2 Report

1  In introduction section, please add some quantitative descriptions for the development of circular dichroism, since this version is all qualitative descriptions.

2  The results are not detailed and comprehensive. Please add more values and larger ranges for your main variables in simulation. And I strongly suggest you to adopt table to present simulation parameters and results.

3  All of the X-label, Y-label and data in figures should have a unit, except for normalized parameter (eg. reflectivity). For instance, in Figure 2, the units of CD and transmittance might be %. And the unit of electric field intensity in Figure 5 might be a.u. (arbitrary units). Besides, please add scale length in Figure 3 and 5 since the nanostructures are specific.

4 The CD values in Table 1 should be adjusted to the same unit. And the unit should appear in table header only. For example, Wavelength range (nm) and CD value (%).

Reviewer 3 Report

The authors have responded to my comments and I recommend the publication of the paper in the present form.

Author Response

Thank you very much for your work and consideration in the publication of our paper. On behalf of my co-authors, we would like to express our sincere gratitude to you.

Round 3

Reviewer 2 Report

Most revised suggestions are adopted. And a minor revision should be completed before publication.

1   Add quantitative descriptions for the development of circular dichroism in introduction section.

2   Check the CD values in Table 1. For example, the CD value should be 18% rather than 0.18% in Ref.24.
